# Exploiting the Native Microorganisms from Different Food Matrices to Formulate Starter Cultures for Sourdough Bread Production

**DOI:** 10.3390/microorganisms11010109

**Published:** 2022-12-31

**Authors:** Natali Hernández-Parada, Oscar González-Ríos, Mirna Leonor Suárez-Quiroz, Zorba Josué Hernández-Estrada, Claudia Yuritzi Figueroa-Hernández, Juan de Dios Figueroa-Cárdenas, Patricia Rayas-Duarte, María Cruz Figueroa-Espinoza

**Affiliations:** 1Tecnológico Nacional de México/Instituto Tecnológico de Veracruz, M.A. de Quevedo 2779, Col. Formando Hogar, Veracruz C.P. 91897, Mexico; 2CONACYT-Tecnológico Nacional de México/Instituto Tecnológico de Veracruz, Unidad de Investigación y Desarrollo en Alimentos, M.A. de Quevedo 2779, Veracruz C.P. 91897, Mexico; 3Centro de Investigación y de Estudios Avanzados del IPN (CINVESTAV Unidad Querétaro), Libramiento Norponiente 2000, Fracc. Real de Juriquilla, Querétaro C.P. 76230, Mexico; 4Robert M. Kerr Food & Agricultural Products Center, Oklahoma State University, 123 FAPC, Stillwater, OK 74078-6055, USA; 5Qualisud, Univ Montpellier, Avignon Université, CIRAD, Institut Agro, IRD, Université de la Réunion, F-34398 Montpellier, France

**Keywords:** sourdough, starter cultures, food microorganisms, lactic acid bacteria (LAB), yeast

## Abstract

The use of sourdough for bread production involves fermentation, which is dominated by lactic acid bacteria (LAB) and yeast. Sourdough can be inoculated with a starter culture or through a food matrix containing microorganisms to initiate sourdough fermentation. Sourdough is used as leavening agent for bread making, and metabolites produced by LAB and yeast confer a specific aroma and flavor profile to bread, thus improving its sensory attributes. However, few publications report the effect of microorganisms from different food products and by-products on sourdough fermentation. This review focuses on using different starter cultures from various food sources, from wheat flour to starter cultures. Additionally, included are the types of sourdough, the sourdough fermentation process, and the biochemical transformations that take place during the sourdough fermentation process.

## 1. Introduction

Sourdough has been used traditionally in the bakery industry as a leavening agent since ancient times; this is achieved via a sourdough starter made with a mixture of flour and water naturally fermented by lactic acid bacteria (LAB) and yeasts [1,2,3,4,5].

During sourdough fermentation, LAB, and yeasts, along with endogenous enzymes, are responsible for heterogeneous microbial metabolism and enzymatic reactions of carbohydrates, phenolic compounds, lipids, and proteins. Carbohydrate metabolism influences the texture, water-holding capacity, shelf life, nutritional factors, and overall flavor of the bread. Phenolic compounds possess antioxidant activity and lipid metabolism can contribute to the production of volatile compounds. Proteolysis increases gluten solubility and its susceptibility to enzymatic degradation, liberating peptides and amino acids with potential physiological properties such as antioxidant, anticancer, and antihypertensive properties [6,7].

Abundant evidence in literature reports support that sourdough fermentation improves the sensory and rheological properties of bread dough, increases bread quality in terms of texture, volume, flavor, and shelf life, delays aging, protects bread against microbial contamination as well as produces pleasant aroma compounds [8,9]. Similarly, it has been shown that the application of sourdough in bread making improves the nutritional value of foods by decreasing the glycemic index and sodium content, increasing the bioavailability of minerals, and promoting the production of bioactive compounds [10,11].

On the other hand, it has been demonstrated that the use of fermentative consortia in sourdough fermentation processes influences its sensory, nutritional, and functional properties [11]. Starter culture for sourdough refers to the microorganisms used to conduct fermentation to increase the yield or to obtain specific dough attributes [12]. However, the ability to adapt to the sourdough ecosystem seems to be one of the essential characteristics for selecting microbial strains [13]. Several studies report the use of autochthonous isolates as starter cultures; however, microbial strains isolated from other food systems, such as yogurt, kefir, and other food sources, are also commonly used [9,14,15,16,17,18]. This review aims to overview sourdough fermentation, focusing on the use of different starter cultures from several food matrices.

## 2. Bread

Bread is a staple food consumed worldwide. Traditional yeast-bread is made from wheat flour; however, it is possible to use other cereals to obtain different kinds of flour for bread making. Wheat flour proteins form a viscoelastic dough during mixing water and flour, that during fermentation and baking, produce bread with a spongy texture. The bread characteristics are due to physicochemical and biochemical transformations that occur during the kneading and fermentation of dough and to the chemical reactions that take place during the bread baking [19,20].

### 2.1. Types of Bread

Bread can be classified according to the fermentation method used for its production, which can be fermented using *Saccharomyces cerevisiae* commercial yeast (instant, active dry, or fresh) or naturally with a starter dough, commonly known as sourdough [21,22].

#### 2.1.1. Commercial Bread

Industrially produced bread is made with baker’s yeast *Saccharomyces cerevisiae*. Bread formulation includes wheat flour, salt, yeast, and water; sometimes, adjuvants are added to improve their characteristics. The fermentation of baker’s yeast produces aroma compounds, CO_2_, and alcohol by consuming the carbohydrates from the starch in wheat flour. In commercial bread fermentation takes approximately 1 to 2 h to obtain the leavened products with pleasant sensory characteristics. For example, crusty or artisan fresh bread usually has an attractive brown, crisp crust, a pleasing aroma, acceptable slicing characteristics, and a soft, elastic crumb texture; while soft crust breads, the most abundant worldwide have a foamy delicate texture [20,21].

#### 2.1.2. Sourdough Bread

Sourdough is a natural starter for bread fermentation. It is one of the oldest fermentation processes used for bread production in different world regions [6]. The sourdough bread making process can take up to two weeks for the starter to mature. Therefore, commercial yeast has replaced the sourdough preparation technique to avoid its laborious preparation and reduce the fermentation time. Although the fermentation time is reduced to 2 h, in the case of commercial baking, more complex aroma and flavor precursor compounds are obtained during the long fermentation process of sourdough bread. For this reason, there is interest in recovering the aromatic and sensory properties of a well-formulated sourdough and its positive effect on the technological, nutritional, and rheological properties of bakery products made with sourdough [12].

## 3. Wheat Flour

Wheat flour is obtained from the industrial milling process of wheat grain. Its structure is shown in Figure 1. Milling removes the protective outer layers of the grain know as wheat bran, representing approximately 14% of the weight of the grain. The bran has high content of fiber and minerals (ash). The germ or the embryo of the wheat grain, makes up only 3% of wheat and contains most of the essential lipids and nutrients. Finally, the endosperm is about 83% of the wheat grain and has a high starch content. It also contains proteins such as gliadin and glutenin, which constitute gluten, after hydration and kneading. The starchy endosperm is reduced to white flour. It is not homogeneous because a protein gradient is present within the starchy endosperm, with the outer layer richer in proteins compared to the inner layers. In addition to being the major component of wheat, the endosperm provides energy and protein for the development of the new plant and is the main constituent of wheat flour. Cells in the outer layer of the endosperm form a barrier between the endosperm and the wheat bran; this layer, called aleurone, is about 6.5% of the grain. It is more biologically active than the endosperm due to its high enzyme content. In addition, the aleurone can affect wheat flour’s functionality and quality; however, during milling operations, this layer is removed, so it is part of the wheat bran [22,23,24].

Wheat flour has a pH between 6.2–6.5 and is composed of water (14%), starch (70–75%), protein (10–12%), non-starch polysaccharides (2–3%), and lipids (2%) [26]. Starch is composed of two glucose polymers, amylose and amylopectin. Amylose is composed of linear chains of glucose linked by α-(1,4) bonds with a helical structure. Amylopectin is a branched molecule constituted of linear chains of α-(1,4)-linked glucoses and branched through α-(1,6) bonds [26,27,28]. Amylose and amylopectin accounts, respectively, for 20–30% and 80–70% of the starch fraction [24,29]. This polysaccharide contributes to different characteristics of wheat-based foods, such as moisture retention, viscosity, texture, flavor, and shelf life [30].

The lipid content in the wheat endosperm is about 2%, most of which comes from cell membranes and the starchy endosperm. Lipids are essential in baking because they influence bread volume and crumb by stabilizing the carbon dioxide gas produced during fermentation and oven spring expansion during baking. However, lipids in the dough can degrade to fatty acids and then oxidize, producing unpleasant aromas. In wheat flour, there are three lipid types: free, bound, and starch lipids. Most free lipids are non-polar, but when the flour is hydrated during dough preparation, they are partially bound. On the other hand, free and bound lipids are called surface lipids since they are outside the starch granule and have an essential role in the functions of the gluten network and the rheological characteristics. Starch lipids are bound to amylose (forming inclusions inside the helical structure of amylose of starch granules), so they cannot react with proteins during dough mixing. However, they can affect the technological properties of starch [24,31,32].

Wheat flour proteins are grouped based on their solubility in different solvents as described by the Osborne’s system; albumins are water-soluble proteins, globulins are soluble in saline solution, prolamins are soluble in an alcohol solution, and glutelins are acid or alkali-soluble. Table 1 shows the range of protein composition of wheat flour [33,34,35].

Gluten is formed during the hydration of the complex mixture of wheat storage proteins (about 80–85% protein) consisting mainly in gliadins and glutenins. The gluten network determines the viscoelastic properties of dough. Glutamine, proline, and cysteine are the predominant amino acids in gluten proteins. The sulfur side group of cysteine residues on gluten protein structures have a sulfur-containing side group that helps to initiate thiol-disulfide exchange reactions allowing building the three-dimensional gluten network during mixing [36,37].

The elasticity and strength of the gluten protein network is provided by glutenin subunits crosslinked together by intramolecular disulfide bonds that entangle with gliadins to form a viscoelastic network [38,39]. The gluten structures are stabilized by forming hydrogen bridges, whereby, when stress is applied, the hydrogen bridges break, making the molecule unstable. Once the stress is removed, the molecule returns to its stable conformation; this ability to restructure itself under stress explains the elastic nature of glutenin [24,37]. On the other hand, gliadins contribute to flow properties conferring extensibility to the dough. Gliadins are a family of proteins with similar amino acid sequences classified into α, β, γ, and ω according to their electrophoretic mobility. α-, β-, and γ- gliadins contain 3 to 4 disulfide bonds from 6 to 8 cysteine residues which moderately disrupt the disulfide bonds between glutenins conferring extensibility to the dough [37,39,40,41]. For their part, albumins and globulins content in wheat proteins accounts for 10–25% and are commonly found in the embryo and aleurone layer. These proteins may act as a nutrient reserve for embryo germination, influence grain hardness, and are enzymes and enzyme inhibitors. The most common albumins and globulin proteins are α-amylase/trypsin, serpins, and purothionins (for a review see reference [42]) [24,42].

Non-starch polysaccharides present in flour are commonly arabinoxylans and β-glucans. Arabinoxylans are formed by arabinose and xylose molecules linked with α-(1,4) bonds. The alcohol moiety of some arabinose residues can be esterified by ferulic acid. Arabinoxylans solubility is related directly related to its molecular size and inversely proportional to the number of arabinose side chains [24]. For their part, β-glucans are composed only of glucose molecules linked with β-(1,3) and β-(1,4) bonds without branching. Both polysaccharides are found in the starchy endosperm and aleurone cell walls and influence the flour hydration by altering starch adhesiveness characteristics; this is because 1/3 of the water in the dough is bound to these polymers [23,24,43].

## 4. Sourdough

Sourdough is a mixture of flour and water fermented by a complex microbiota that includes lactic acid bacteria (LAB) and yeasts [2,3,26]. From a microbiological perspective, sourdough is a stressful ecosystem for microorganisms whose metabolites cause dough acidification and leavening in addition to aroma and flavor compounds [26]. Sourdough acts as a leavening agent in bread making; it yields a more aromatic bread with better texture and flavor, extended shelf life, and nutritional benefits due to the presence of nutrients such as amino acids, vitamins, minerals, and dietary fiber [26,44,45].

Fresh starter sourdoughs are characterized by their high availability of carbohydrates that initiate the fermentative process, where the microorganisms are not yet well established. For this reason, fresh starter is generally not used as a leavening agent for bread and lacks complex flavors. It possesses a floury, bland, and flat taste, with a floury and inactive smell. As the fermentative process progresses, active starter sourdoughs are obtained, with large CO_2_ bubbles and a fermented, sweet, sour, and not floury taste, accompanied of a slightly sour and milky sweet smell, resulting from sequential step fermentation [3,32]. Mature starter sourdoughs are obtained, which are rich in CO_2_ but deficient in nutrients for microorganisms and are characterized by an excess of sour and strong vinegary taste, and a strong, vinegary, pungent, and fermented smell, tiny gas bubbles, and a collapsed structure, as shown in Figure 2 [3].

### 4.1. Sourdough Fermentation

Sourdough fermentation usually occurs under limited aerobic conditions and involves a succession of LAB and yeasts. Cereals with pH values between 5.0–6.2 and high concentrations of fermentable carbohydrates are suitable sources for the growth of LAB strains until the dough pH reaches an approximate value of 4.0. After that, acid-tolerant yeasts are the predominant microorganisms in the fermentation process [3,46,47,48].

#### 4.1.1. Lactic Acid Bacteria (LAB)

Lactic acid bacteria (LAB) are the predominant microorganisms in sourdough fermentation with a cell concentration of 10^8^–10^9^ CFU/g and are responsible for sourdough acidification [49]. LAB species contribute to the dough acidification process; however, heterofermentative species also contribute partially to the leavening process and are of greater importance in sourdough than homofermentative LAB species (commonly used in other fermented food products). The metabolism of heterofermentative LAB produces lactic acid, CO_2_, acetic acid, and ethanol by heterolactic fermentation using glucose as a carbon source. These metabolites decrease dough pH at values lower than 4.5 [50,51]. The relationship between the lactic acid and acetic acid produced during sourdough fermentation is an important parameter known as fermentation quotient (FQ). It indicates the molar ratio of lactic acid/acetic acid and is calculated as FQ = (g of lactic acid in 100 g of dough/molecular weight of lactic acid):(g of acetic acid in 100 g of dough/molecular weight of acetic acid). The FQ should be between 2.0 and 2.7 to achieve a good bread flavor. This parameter is related to the type of LAB (homo and hetero-fermentative) that dominates in the fermentation, and its value influences the sensory properties of sourdough breads [52,53,54].

During sourdough fermentation, LAB moderately hydrolyzes starch, performs proteolysis, and acidifies the dough, obtaining soft and pleasant tasting crumb, increasing mineral bioavailability through phytate degradation, and preventing the growth of spoilage microorganisms. The proteolytic activity of LAB releases amino acids and peptides, which stimulate their growth and synthesis of B-complex vitamins and volatile fatty acids, which provide better nutritional conditions for increasing yeast counts [55]. LAB species found naturally in sourdoughs are commonly from the genera *Lactobacillus* (*L. crispatus*), *Lactiplantibacillus (Lpb. plantarum)*, *Fructobacillus (F. sanfranciscensis* formerly *L. sanfranciscensis)*, *Levilactobacillus (Lev. brevis)*, and *Limosibacillus (Lim. pontis*). Furthermore, other species of the genera *Leuconostoc*, *Weissella*, *Pediococcus*, *Lactococcus*, *Enterococcus*, and *Streptococcus* were isolated from the sourdough [50,51,56].

#### 4.1.2. Yeast

Yeasts are present in sourdough fermentation in cell concentrations between 10^6^–10^7^ CFU/g and are responsible for the leavening action of sourdough. The ratio of LAB and yeast is generally 100:1. Furthermore, yeasts produce various aromatic compounds that contribute to harmonious flavors in bread in combination with acids. On the other hand, yeasts present in sourdough show adaptation to stress environments created by low pH values, high carbohydrate concentrations, and high LAB cell density [57,58]. Yeasts improve the bread flavor by producing metabolites that confer flavors, such as esters, aldehydes, and acetoin. Other compounds produced by yeasts, like glutathione, glycerol, and pyruvic acid, contribute to the textural structure of bread by enhancing the gluten network [59]. The most reported yeast species in sourdough belong to the genera *Saccharomyces*, *Candida*, *Pichia*, *Torulaspura*, and *Rhodotorula*, as shown in Table 2 [60,61,62].

Sourdough microorganisms must present compatible interactions between them, which may promote their growth (and sometimes a symbiotic effect) to remain metabolically active during fermentation. Bacteria and sourdough yeasts partially compete for nitrogen sources. However, yeasts have been shown to produce essential amino acids to facilitate LAB growth in co-cultures [77]. Finally, the most limiting factor for fermentative activity is the lack of substrate, which is solved by sourdough feeding [26,78].

#### 4.1.3. Biochemical Transformations during Sourdough Fermentation

Sourdough fermentation has different biochemical effects on the major compounds of dough as shown in Figure 3.

Fermentable carbohydrates such as maltose, glucose, and maltodextrins are released from the starch degradation by endogenous enzymes such as amylases that release maltodextrins [76,79]. Amylases cannot degrade native starch granules in flour, but when granules are damaged (during the milling process), the partial degradation of starch takes place [75]. Lactic acid bacteria with amylolytic capacity (ALAB) have been reported in different types of foods, mainly starchy [80,81]. The ALAB can produce extracellular amylolytic enzymes for the degradation of starchy substrates. Therefore, these bacteria can directly utilize starch to produce mainly lactic acid [82,83]. The amylolytic enzymes mostly produced by ALAB strains are amylases, amylopullalanases, and pullulanases [84]. The production of these enzymes has been studied with *Lb. amylovorus* NRRL B-4540 [85,86], *Lpb. plantarum* A6 [87], and *Lcb. manihotivorans* LMG18010 [88]. Recently, maltose-forming amylase from *Lpb. plantarum* strain S21 has been characterized [83].

Cereal flour and its microbiota contain proteases and peptidases that hydrolyze proteins during sourdough fermentation [76,89]. The primary proteolysis during sourdough fermentation is due to indigenous flour aspartic proteases, aminopeptidase, endopeptidase, and carboxypeptidase activation is promoted by the drop in pH. Moreover, this acidification increases the solubility of gluten proteins, making them more susceptible to enzymatic degradation [90]. In the second proteolysis, free amino acids are released by the peptidase system of the microorganism [91]. LAB strains contribute to protein degradation; for example, the proteolytic system of *F. sanfranciscencis* contains aminopeptidase, dipeptidase, and a cell wall-associated serine proteinase [92,93]. On the other hand, the adaptation and growth of some sourdough LAB strains depend on flour amino acids, flour proteases, and their proteolytic system to satisfy the nitrogen metabolic requirements. This dependence is because these bacteria lack proteinase in their cell envelope. Therefore, the availability of oligopeptides and amino acids depends on the action of cereal-active proteases under acidic conditions. Similarly, yeasts require amino acids as a nitrogen source fermentation; however, their enzymatic activities allow them to provide glucose and fructose (invertase) and amino acids (protease) to LAB strains [89,94].

Phenolic compounds (phenolic acids, flavonoids, and tannins) in wheat flour are commonly found in bound forms. During sourdough fermentation, the cereal and LAB enzymes release the phenolic compounds [93]. In wheat, the major phenolic acid is ferulic acid (FA), accounting for almost 90% of total phenolic compounds. The remaining 10% corresponds to caffeic acid, dihydrobenzoic acid, and sinapic acids. In sourdough fermentation, FA is transformed into 4-vinilguaiacol (natural aroma of buckwheat), ethyl-guaiacol (smokey flavor), and dihydroferulic acid (antioxidant) by the enzymatic activity of LAB and yeast [95].

Sourdough addition increases the aromatic profile of bread due to LAB and yeast production of diverse volatile organic compounds (VOCs) such as alcohols, aldehydes, acids, ketones, and esters that benefit the flavor and aroma of the final product. LAB produces compounds such as acetaldehyde (buttery), hexanoic acid (fatty), 1-hexanol (ethereal), 1-pentanol (oil), 2-pentylfuran (fruity), and 2-methyl butyl acetate (overripe fruit). For yeast, it has been reported that they mainly produce ethanol (alcohol), 2-methyl-1-propanol (ethereal), 3-methylbutanol (alcohol), 2-phenylethanol (rose), 2-methylbutanol (roasted), aldehyde (acetaldehyde, hexanal), 3-hydroxy-2-butanone (sweet), benzyl alcohol (floral), benzaldehyde (almond-like), 2-phenylethanol (rose), and 2,3-butanedione (buttery) [94]. Recently, Xu et al. [96] selected yeast strains to mix with *F. sanfranciscensis* as mixed starter culture according to the volatile organic compounds produced. *M. guillermondii* EH1 y *P. kudriavzevii* EP1 were the selected strains. When compared to the culture of *F. sanfranciscencis*, both mixed cultures present specific compounds such as ethyl-acetate (sweet, floral), ethyl-hexanoate (fruity, liqueur), isoamyl acetate (banana), propyl lactate (fruity, liqueur), phenyl ethyl acetate (floral). However, with *P. kudriavzevii* EP1, the isoamyl lactate (cream and walnut) was obtained, and with *M. guilliermondii* EH1, the compounds (E, Z)-2,6-nonadienal (cucumber) and ethyl decanoate (fruity).

Furthermore, the release of reducing sugars and amino acids during fermentation also contributes to the formation of aromas and bread crust color through the Maillard reaction during baking [26,97,98]. The presence of the dominant microorganisms in sourdough is influenced by fermentation parameters, such as dough yield, salt, quality, and quantity of starter used, as well as the number of propagation steps, fermentation time, and oxygen availability determine the presence of the dominant microorganisms in the sourdough [3,99].

#### 4.1.4. Fermentation Parameters

Fermentation parameters are essential factors for sourdough maintenance and can affect the initial composition and performance of the starter. The most influential are flour type, hydration, temperature, time, refeeding practices, and sourdough environment [3,100].

Wheat Flour Type

Wheat flour provides different nutrients, mainly carbohydrates and amino acids. Therefore, its composition can influence the growth of microorganisms present during sourdough fermentation [8,101]. Likewise, the ability of the microorganisms to metabolize the carbohydrates present in the dough also affects their development. However, the preference for carbon sources is not the only factor that influences the microorganisms present in dough. The availability of carbohydrates also impacts the organisms present and this is related to the action of amylases in each type of flour. Amylases contribute to the production of fermentable carbohydrates through the degradation of complex carbohydrates; this favors bacterial growth and causes the development of acid-tolerant microorganisms [54,102].

Dough Yield

The dough yield (DY) refers to the amount of water mixed with the flour, i.e., the ratio of flour to water when 100 g of flour are used. If the DY value is between 150–160, the sourdough is considered firm; with DY values between 200–300, the sourdough is semi-liquid, and when the DY is more than 300, the sourdough is liquid [99]. The dough yield index is obtained using the following expression:(1)$$DY=\frac{\left(flour\;mass\left(g\right)+water\;volume\;(\mathrm{mL}\right))}{flour\;mass\;\left(g\right)}*100$$

The presence of water in the sourdough is relevant for the metabolic activity of microorganisms to remain active since water activates proteolytic enzymes [3].

Temperature

The temperature of sourdough fermentation and storage affects the physical, chemical, biochemical, and sensory characteristics of sourdough bread due to the impact on the microbial diversity in the dough. Temperature influences the predominance of bacteria and yeasts, their growth and metabolic activity [103,104]. It is essential to consider the temperature of sourdough fermentation, refeeding and at which the sourdough is added during the baking process. However, the sourdough fermentation temperature includes incubation and cooling temperature during a daily sourdough feeding cycle. The most used temperature for sourdough fermentation is 30 °C [19,71,105].

Fermentation Time

During sourdough fermentation, it is necessary to consider the starter maturity since it determines the species of microorganisms present. In addition, there is a dynamic maturity of sourdough at which a maximum leavening power is obtained followed by a declined as it continues aging [20,87]. The frequency of refeeding is also essential because the diversity of microorganisms in sourdough varies depending on this frequency. There is a range in the feeding frequency that can optimize the fermentative capacity of sourdough since, with the correct frequency, the growth of undesirable microorganisms such as fungi can be suppressed. In addition, if the fermentation time is short (more frequent feedings), the culture will be selective with the species present in the sourdough [3]. The sourdough fermentation time influences the acidification, leavening power, and cell density of the yeasts and LAB in the starter. Longer fermentation times of sourdough with added by-products is attributed to adaptation of microorganisms to the environment of sourdough [80]. Calvert et al. [3] reported an ideal fermentation time at which the sourdough reaches an equilibrium between acidity and microorganisms’ growth; this fermentation time allows for optimizing bread’s leavening and sensory properties. However, the ideal fermentation time is relative because it depends on the microbial strains and the baker’s preference.

Backslopping

Sourdough backslopping is the number of sourdough propagation (refreshing) steps where an amount of the starter is discarded by adding water and flour to the remaining sourdough. There are two types of replenishment: the amount of starter used to refresh the sourdough during fermentation, which is commonly 20% of inoculum, and the amount of sourdough used in baking which influences the aromatic profile and acidity of the bread [3,99].

Environment

The environment (also called “house microbiota”) in which sourdough is elaborated influences its microbial diversity since different species of microorganisms have been reported in sourdough depending on the geographical region and the environment where they develop [3]. However, a recent study of how the microbial diversity of sourdough starters varies across and between continents demonstrates which species of bacteria and yeast are commonly found in sourdough starters and suggests the geographical location has a relatively small influence on the microbial diversity of these cultures [1]. Instead, microbial diversity depends more on starter obtention and maintenance practices [1]. This information coincided with the report by Comasio et al. [106]. This research showed that the microbial sourdough diversity was influenced by the environment microbiota of the producer and the process parameters applied. Despite the artisan producer’s use of different flours, the sourdoughs contained a similar microbial population, independent of the flour used.

Thus, more studies about the geographical dispersion of microorganisms in sourdough along continents are needed to establish if there a relationship between the geographical place of sourdough development and the yeast and LAB dominance during fermentation. Additionally, it remains unclear if an ideal environment for sourdough production exists due to the differences of the house microbiota caused by the artisan practices.

### 4.2. Classification of Sourdough

Four sourdough types are classified concerning the technology used for sourdough production, as shown in Figure 4 [26,94]. The fermentation of each kind of sourdough is characterized by different fermentative microbiota which depends on the maintenance factors [103].

#### Sourdough Types

Type I

Type I sourdough forms a firm dough with a DY between 150–160. This type of sourdough is obtained by fermenting a mixture of wheat flour and water indigenously with the LAB and yeasts present in the flour, the environment, or any food matrix used to initiate fermentation [94,101]. It is characterized by feeding the dough to continuously spread and keep the microorganisms metabolically active [3]. Generally, bakers apply a backslopping step based on three batches fed for 24 h, thus obtaining the dough for baking. In the last stage, sourdough is used as a leavening agent; it can be considered a natural starter culture with different microbial strains [104,107].

Type II

This type of sourdough has a fermentation time of 2 to 5 days at a temperature higher than 30 °C; it has a DY between 200 and 300, so its water content increases compared to type I sourdough, making it a semi-fluid dough. This sourdough involves the inoculation of LAB and yeast in the dough. Because of the high fermentation temperatures, the production of organic acids is favored, which results in a decrease of pH to values below 3.5. This type of sourdough is mainly used for acidification and leavening. This sourdough can be stored for up to one week. Its consistency and formulation allow specific flavor profiles with different bread textures and volumes [94,108].

Type III

Type III sourdough is obtained by dehydrating sourdoughs obtained by traditional fermentation or by adding starter cultures. The drying process can be performed by freeze-drying, spray drying, or drying in a fluidized bed reactor. It is essential to keep in mind that the microorganisms present in the dough must be resistant to drying [105,109,110]. This type of dough has advantages over the other types since it has a longer shelf life, occupies less volume, and is easier to handle transport and store. In addition, with Type III sourdough final products are standardize with consistent bread flavor and aroma [3,111].

Type IV

In this case, the sourdough is initiated with a starter culture (yeast, LAB, or LAB-yeast-starter) that is propagated with traditional backslopping until obtain a mature sourdough. The dough can be firm or semiliquid [12,104]. The growth of the consortium of LAB and yeasts added to this sourdough depends on the type and quality of flour used, fermentation conditions, and interactions with the indigenous microbiota. On the other hand, this sourdough can also be dehydrated [3,99,103].

## 5. Starter Cultures Used for Sourdough

Although the flavor, aroma, and sensory quality of sourdough bread are unique, the use of the sourdough fermentation technique to produce various bakery products represents several challenges for the bakery industry. These are due to the complexity and hand labor needed for the production and maintenance of sourdough through daily refreshments and the long fermentation times. Therefore, the application of starter cultures that simplify and shorten the fermentation process at the commercial level has been proposed [54,112,113,114,115]. Medina-Pradas et al. [116] defined a starter culture as a preparation of live microorganisms used deliberately to accelerate the fermentation, triggering specific changes in the food substrate’s composition and sensory properties to get a more homogeneous product. Suitable selection of the microorganisms to be used as starter cultures is of utmost importance for sourdough application, considering that it significantly influences the final dough’s characteristics, pH, and fermentation temperature.

Furthermore, several studies have shown that using autochthonous LAB for sourdough fermentation represents a biotechnology tool for exploiting the potential of non-wheat cereals in bread making [115,117,118,119]. In recent years, sorghum has gained increasing attention worldwide, especially in Western countries, due to its valuable nutritional quality and health-promoting components. Furthermore, sorghum is gluten-free, making it an alternative food for patients with celiac disease [6,115,118,120]. On the other hand, it has also been reported that the use of new starter cultures, including different *Weissella* species, to produce sourdough bread [120,121] improves the rheological properties of the dough due to the production of exopolysaccharides (EPS) [115,122].

During sourdough production, starter cultures are used for elaborating sourdough types II, III, and IV. Type II sourdough is made by a process known as an industrial method, which consists of a single-stage fermentation with LAB culture or mixed culture (LAB with yeast) for 15–24 h and then backslopped stages [104,123]. The LAB and yeasts commonly used in sourdough Type II belong to species *S. cerevisiae*, *Lb. amylovorus, Lim. panis, Lim. fermentum, Lev. brevis*, *Lim. pontis*, and *Lim. reuteri.* These bacteria are characterized by their thermotolerance and acid tolerance [104,124,125,126]. It was reported that type II sourdough has several advantages over type I, such as a single fermentation stage, improved control of fermentation parameters (pH, temperature, acidity), and easier nutrient addition, which results in enhanced performance and control of microbial metabolism [86]. Therefore, the risk of mold contamination during fermentation is reduced by accelerating the process. In addition, sensory properties are increased, and final products are standardized due to the selection of the starter culture and the further production of important metabolites. All these characteristics make type II sourdoughs suitable for use in industrial processes. Nevertheless, due to the complex microbiota of sourdough, a critical step is the selection of the strains to be used for the starter culture [104]. Gaggiano et al. [127] proposed a protocol for producing and using a defined, multi-specific, semi-liquid sourdough starter culture to meet industrial requirements.

For the case of type III sourdough, produced by dehydrating the stabilized form of type II sourdough, it is essential to ensure the selection of the starter culture based on its ability to rapidly acidify the flour-water mixture and/or its ability to produce specific flavors [123]. Since some companies commercializing type III sourdough do not verify the viability of the sourdough microbiota and only consider the texture and aroma enhancement of the final products, it is necessary the addition of commercial yeast to allow leavening. In this respect, to ensure a stable starter culture for this type of sourdough, there is a need to consider the stability of the starter culture during the drying process. Some examples of LAB strains resistant to drying are *Lactiplantibacillus plantarum* Ls71, *Pediococcus acidilactici Ls72*, and *Lentilactobacillus buchneri* Ls141 [128].

The sourdough type IV is typically used in laboratory studies and some artisanal bakeries. Starter cultures or another inoculum such as fruits or honey can be added, and fermentation takes longer than Type II because of the backslopping techniques. Competition between the added starters and the autochthonous strains species can exist. The more competitive or well-adapted strains may establish their dominance, and a natural selection will occur [104,125]. Therefore, selecting strains that dominate the environmental conditions and drive sourdough fermentation is essential to a set of desired characteristics [94]. Commonly the LAB and yeasts species whose use has been reported as starter cultures for this type of sourdough belong to *Lim. fermentum*, *Lpb. plantarum*, *F. sanfranciscensis*, *Lc. mesenteroides*, *W. confusa*, *S. cerevisiae*, *C. humilis*, and *Wickerhamomyces anomalous* [103].

### 5.1. Traditional Starter Cultures

As mentioned before, the main source of the microorganisms used for traditional sourdough production is flour and water mixture used for fermentation [14].

Paramithiotis et al. [129] studied the metabolic interactions between the dominant species of traditional Greek wheat sourdough starter *F. sanfranciscensis* and *S. cerevisiae* and the accompanying microbiota *Lvb. brevis*, Co. *paralimentarius*, *Ped. pentosaceus* and *W. cibaria*. In this study, *F. sanfranciscensis* and *S. cerevisiae* were used as starter cultures, alone or in combination with *Lvb. brevis*, Co. *paralimentarius*, *Ped. pentosaceus* and *W. cibaria*. Metabolic products were determined in the sourdough samples by HPLC analysis. The results showed that *Lvb. brevis*, *W. cibaria* and *Ped. pentosaceus* had basically no effect on the growth of the main microorganisms or on total metabolite production. In contrast, *Co. paralimentarius* showed a negative effect on the growth of *F. sanfranciscensis*. All sourdough breads produced had suitable sensory properties. The bread made with *S. cerevisiae*, *F. sanfranciscensis* and *Lvb. brevis* ranked first in the sensory evaluation.

In recent years, the role of these autochthonous microorganisms, mainly yeasts and LAB, in functional and technological properties has been evaluated, some of these studies are mentioned below.

Sidari et al. [130] evaluated a mixed starter culture for sourdough bread production. This culture consisted of *F. sanfranciscensis* B450, *Lc. citreum* B435 and *C. milleri* L999. The researchers assessed the viability of the strains during sourdough production in the laboratory until production in the bakery plant, as well as the effect of the starter culture on the antioxidant and rheological properties of the sourdoughs and the resulting bread. In this work, the viability of *F. sanfranciscensis* B450 and *C. milleri* L999 was demonstrated. One of the sourdoughs inoculated with this starter culture (PF7 M) had a higher phenolic content and antioxidant activity (measured by DPPH) than the artisan bakery sourdough. The other sourdough (PF9M) was shown to have an increase in texture parameters.

Fekri et al. [131] isolated and selected yeasts and LAB from traditional Irani sourdough. The selection criteria used were phytate degradation ability, tolerance to acidity, and bile salts. The selected microbial strains were used to produce sourdough bread and were compared with a yeast-bread (inoculated with *S. cerevisiae*). Several nutritional parameters (such as antioxidant activity, EPS production, phytic acid content, and in vitro starch digestibility) and technological parameters (bread quality and sensory test) were evaluated during the study. The microorganisms used as starters for sourdough fermentation were *Kluyveromyces marxianus*, *K. lactis*, *K. aestuarii*, *E. faecium*, *Ped. pentosaceus*, and *Lc*. *citreum*. It was found that the sourdough bread inoculated with *Kluyveromyces aestuarii* had the highest sourdough porosity (70.4%), the lowest hardness (508.7 g), the highest concentration of EPS, and favorable sensory attributes.

Boyaci-Gunduz et al. [132] identified LAB strains from sourdough samples of different cities in Turkey with culture-independent and culture-dependent molecular methods. Thirteen LAB species were identified, mainly *F. sanfranciscensis* and *Lpb. plantarum*. In monoculture and double culture, these species were studied as starters for sourdough production. The chemical and microbiological properties, as well as the VOC profile of the sourdoughs, were evaluated. It was found that the sourdoughs inoculated with monoculture and double culture of *F. sanfranciscensis* RL976 were characterized by higher microbial growth, titratable acidity, lactic acid concentration, and a more significant number of VOCs than other analyzed samples. These characteristics may be essential to ensure the reproducibility and stability of industrial sourdough bread production. Furthermore, the authors highlight that the results of this study corroborate the hypothesis that strains isolated from the sourdough environment are the most promising candidates for developing starter cultures. Therefore, *L. plantarum* and *F. sanfranciscensis* could be applied as dual starter cultures in industrial sourdough production to reach the desired level of acidification and aroma in a short time.

### 5.2. Starter Cultures Formulated with Other Food Matrices Microorganisms

As previously mentioned, native microorganisms from food matrices different from sourdough could be used as starter cultures for this process (Table 3). They could provide the sourdough with interesting metabolic properties and, therefore, modify the physico-chemical and sensory properties of the bread produced with such dough. Moreover, this selection of sourdough starter cultures needs not be limited to LAB and yeast. Still, other microorganisms, such as acetic acid bacteria (AAB) or even some coagulase-negative staphylococci (CNS) could also be used [14].

Graça et al. [16] investigated the effect of incorporating yogurt as a starter in sourdough wheat bread on technological and nutritional properties. In this study, two bread dough matrices were made: endosperm wheat flour (white flour) and blended with whole-grain flour. In addition, two fermentation methods were performed, two-stage sourdough bread and yeast bread fermentation. It was observed that yogurt-sourdough, compared to yeast-sourdough, promoted significant changes in chemical composition, such as a higher degree of protein proteolysis, increased peptide, and free amino acid content (FAA), solubilization of phenolic compounds (46–53%), increased DPPH (2,2-diphenyl-1-picryl-hydrazyl-hydrate) radical scavenging (54–65%) and ferric reducing power (85–88%), especially when whole wheat flour was combined with white wheat flour. Adding yogurt as a baking ingredient for sourdough enhanced bread crumb softness (15–12%) and retarded staling (40–35%). Furthermore, the glycemic index was decreased (18–32%), while there was an enhancement in protein digestibility (6–12%) and bioavailability of free amino acids (50–100%). For these reasons, adding yogurt to sourdough fermentation is a promising alternative to improve wheat bread’s technological and functional properties.

Korcari et al. [71] studied the use of previously selected starter cultures to obtain spelt-based sourdough bread with improved technological, sensory, and preservation properties. They established two consortia, containing one yeast strain (a commercial *S. cerevisiae* strain or a maltose-negative *Kazachstania unispora* strain) and two LAB strains, *Weissella cibaria* and *Pediococcus pentosaceus*. These microorganisms, except for *S. cerevisiae*, were previously isolated in the spontaneous fermentation of corn bran by Decimo et al. [135]. The ability to grow in co-culture, without inhibition between LAB and yeasts, was investigated, which grew in proportions considered suitable for sourdoughs. The performance of the two consortia was evaluated in a spelt-based sourdough bread, and the leavening behavior, crumb softness, bread volume, shelf life, and consumer preference were assessed. The product obtained with the consortium containing *S. cerevisiae* had a superior crumb texture that was maintained during five days of storage and was well accepted by consumers. In addition, both consortia improved shelf life by preventing the growth of common cereal-contaminating fungi. The results indicated that the selected starter cultures have a promising potential to improve the baking quality of products obtained with flours with poor technological performance but interesting nutritional properties [71].

Limbad et al. [133] investigated coconut water kefir (CWK) as a starter culture for wheat-sourdough fermentation. These inoculums consisted of *Lim. fermentum* (at 8.30 log CFU/mL or 4.90 log CFU/mL) and *Lpb. plantarum* (9.60 log CFU/mL), and baker’s yeast can be added. CWK-sourdough fermentations were conducted for 18 or 24 h. Subsequently, they performed physicochemical analyses (shelf life, texture, carboxylic and amino acids profile, and proximate composition of CWK-sourdough bread). The sample of bread inoculated with *Lpb. plantarum* at 9.60 log CFU/mL without yeast and fermented during 24 h produced a higher concentration of organic acids (lactic, succinic, acetic, and pyruvic acids) and amino acid and improved overall bread quality in terms of flavor, shelf life, and texture.

Comasio et al. [106] used several microorganisms derived from different food matrices, such as cocoa bean fermentation, fermented sausage, and water kefir, as starter cultures for sourdough fermentation. The microorganisms studied for Type II wheat sourdough included LAB, AAB, and CNS strains. They studied the microorganisms prevalence in the sourdough ecosystem during 72-h fermentations. *Lim. fermentum* IMDO 222 (cocoa bean fermentation) and *Lat. sakei* CTC 494 (fermented sausage) were able to survive during Type II sourdough productions and seem to be promising candidates as sourdough starter culture strains.

Păcularu-Burada et al. [18] used milk (MKG) and water (WKG) kefir grains as starter cultures for gluten-free sourdoughs (quinoa, chickpea, okara, and buckwheat). They formulated three artisanal starter cultures, two from water (WKG1, WKG2) and one from milk (MKG) kefir grains. The authors studied the combined effects of ingredients, sterilization, gelatinization, and type of fermentation (liquid and solid fermentation) on the biochemical performance of the microbial consortium and its antifungal, antioxidant, and acidification potential. Results demonstrated the potential of WKG as starter cultures to produce gluten-free sourdough. Mainly, inoculation with WKG2 (0.20% by weight) using liquid and solid fermentation was shown to have a higher composition of organic acids, flavonoids, and polyphenolic compounds with antioxidant and antifungal properties.

Yu et al. [64] investigated the effect of using pear and orange as starters for sourdough fermentation on white pan breads, focusing on their acidification capacity, fermentability, free amino acid (FAA) concentration, and bread properties. This work demonstrated that adding sourdough improved bread’s technological and nutritional properties and that different sourdough substrates used as starters resulted in differences in bread characteristics. Breads made with sourdoughs that started with fruit (pear or orange) had lower pH and acidity than control breads. Gas production decreased with the addition of sourdough, but gas holding capacity increased significantly. Sourdough fermentation offered a suitable acidic environment to improve the metabolic activity of some endogenous enzymes, which resulted in an increase in specific volume and FAA concentration. The authors reported isolating 15 yeasts and 26 LAB from the pear-initiated sourdough and 21 yeasts and 18 LAB from the orange-initiated sourdoughs. LAB found in these doughs were identified as *Lvb. brevis*, *Lpb. plantarum*, and *Flb. rossiae*, while yeasts were identified as *S. cerevisiae*.

Choi et al. [134] evaluated two LAB isolated from kimchi, *Leuconostoc citreum* HO_12_ and *Weissella koreensis* HO_20_, as starter cultures for the production of whole wheat sourdough bread. These LAB strains isolated from kimchi were evaluated as starter cultures in whole wheat sourdough bread making. After 24 h of fermentation at 25 °C, both bacteria reached a total count of 9 log CFU/g dough, and the two doughs showed similar pH and total acidity. Sourdoughs and bread with 50% (*w*/*w*) sourdough produced with the starter cultures showed a steady capacity to retard the growth of fungi and bread spoilage bacteria such as *Penicillium roqueforti*, *Aspergillus niger* and *Bacillus subtilis*. It appears that both LAB strains possess the potential to improve the shelf life of wheat bread.

## 6. Conclusions

Different plant or animal origin food matrices, fermented or not, have great potential to be used as starter cultures for the sourdough bread process. Their native microorganisms mainly constituted by lactic acid bacteria, yeasts, and even acetic acid bacteria, positively impacts sourdough fermentation and the overall sourdough bread quality. The field of sourdough fermentation gains interest every day. The use of different food matrices as starter cultures for sourdough fermentation opens a multitude of possibilities to diversity bread making practices and to improve physicochemical, sensory, and nutritional properties of sourdough bread.

## Figures and Tables

**Figure 1 microorganisms-11-00109-f001:**
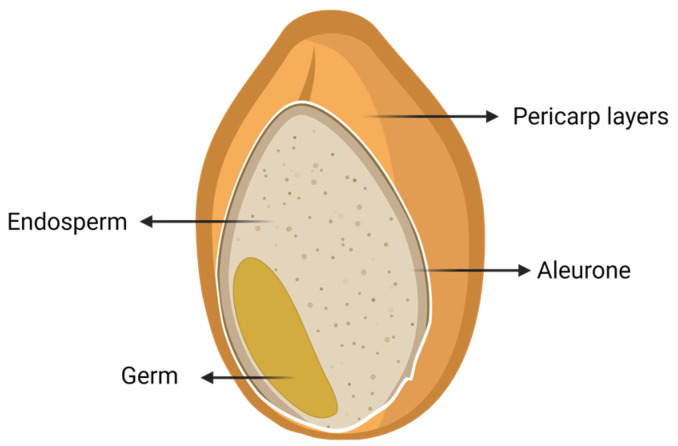
Cartoon depicting the **whole-grain** wheat structure. Adapted from [25]. Created with BioRender.com.

**Figure 2 microorganisms-11-00109-f002:**
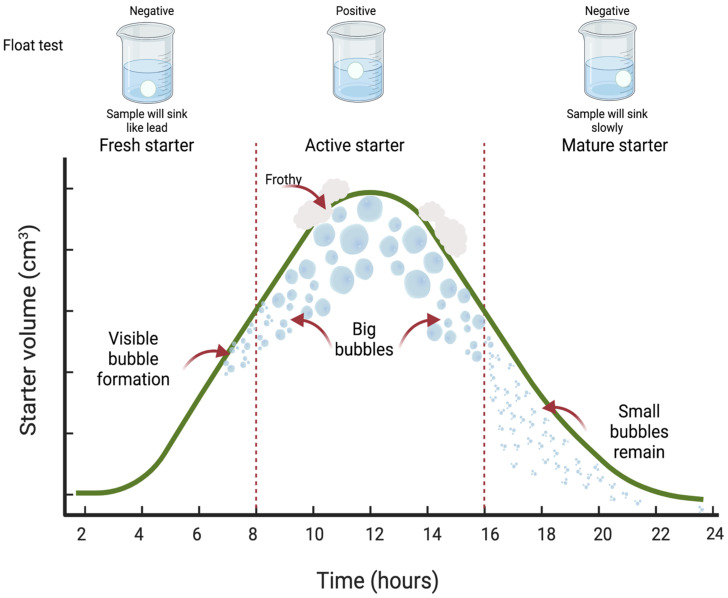
Sourdough starter maturation process during fermentation. Adapted from [3]. Created with BioRender.com.

**Figure 3 microorganisms-11-00109-f003:**
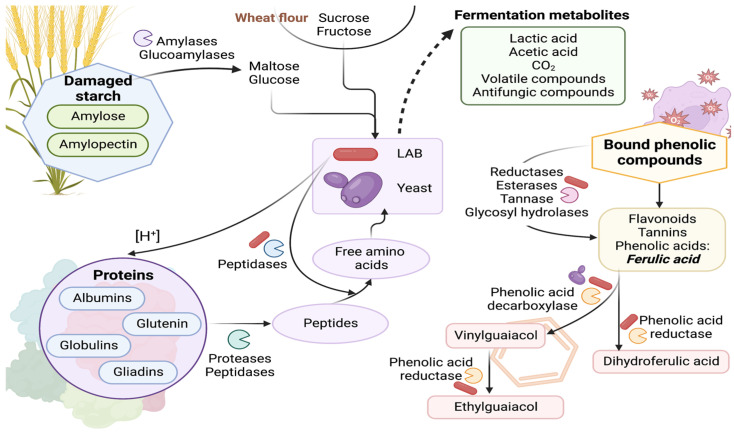
Sourdough biochemical transformations during fermentation [26,75,76,79,80,81,82,83,84,85,86,87,88,89,90,91,92,93,94,95,96,97,98]. Created with BioRender.com.

**Figure 4 microorganisms-11-00109-f004:**
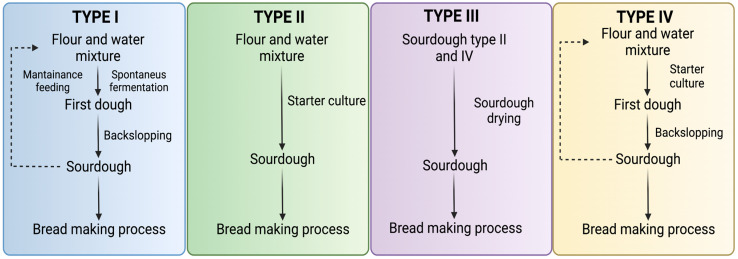
Types of sourdough are classified according to the method applied to obtain them. Adapted from [26,94]. Created with BioRender.com.

**Table 1 microorganisms-11-00109-t001:** Wheat flour protein composition.

Protein	Percentage (% *w*/*w*)
Albumins	5–15
Globulins	5–10
Prolamins (gliadins)	40–50
Glutelins (glutenins)	30–40

**Table 2 microorganisms-11-00109-t002:** Overview of LAB and yeast found in sourdough elaborated with wheat flour.

Country/ Region	Sourdough Type *	Lactic Acid Bacteria (LAB)	Yeast	References
Japan /Asia	Type II	*Lc. citreum, L. lactis*, *W. confusa*, *W. cibaria*, *Lpb. plantarum*, *Lpb. paraplantarum*, *Lvb. brevis*	*W. anomalus*, *Ks. unispora*, *S. cerevisiae*	[63]
China /Asia	Type II	*Lat. curvatus*, *Ped. pentosaceus*, *Lvb. brevis*, *Lpb. plantarum*, *Lc. mesenteroides*, *Flb. rossiae*	*S. cerevisiae*	[64]
China /Asia	Type I	*F. sanfranciscensis*, *Lim. pontis*	*S. cerevisiae*, *C. humilis*	[65]
Italy /Europe	Type I	*Lvb. brevis*, *Lpb. plantarum*, *Lcb. rhamnosus*	*W. anomalus*, *S. cerevisiae*, *T. delbruekii*, *P. kluyveri*, *C. boidinii*, *C. diddensiae*	[62]
Japan /Asia	Type I	*Lvb. brevis*, *Co. alimentarius*, *Lpb. pentosus*	*S. cerevisiae*, *C. humilis*	[66]
France /Europe	Type I	*F. sanfranciscensis*, *Lpb. plantarum*, *Co. kimchi*, *Lat. sakei*, *Lev. hamesii*, *Lpb. pentosus*	*K. bulderi*, *C. humilis*, *K. unispora*, *T. delbruekii*, *R. mucilaginosa*, *C. carpophila*, *S. cerevisiae*, *H. pseudoburtonii*	[67]
Turkey /Asia	Type I	*Lpb. plantarum*, *F. sanfranciscensis*, *Lev. spicheri*, *Flb. rossiae*, *Lev. namurensis*, *Lev. zymae*, *Lcb. casei*, *Co. mindensis*, *Lb. acetotolerans*, *Co. farciminis*, *Co. paralimentarius*, *Ped. pentosaceus*, *E. durans*, *E. faecium*, *Lc. mesenteroides*, *W. confusa*	*S. cerevisiae*, *P. guillermondii*, *T. delbrueckii*, *C. parapsilosis*, *C. pararugosa*	[68]
China /Asia	Type I	*F. sanfranciscensis*, *W. cibaria*, *Lim. fermentum*, *Lpb. plantarum*, *Lim. pontis*, *Co. paralimentarius*	*S. cerevisiae*, *C. humilis*, *W. anomalus*	[69]
Turkey /Asia	Type I	*W. viridescens*, *Ped. pentosaceus*, *Ped. acidilactici*, *Lvb. brevis*, *Len. parabuchneri*	*S. cerevisiae*, *P. membranifaciens*	[70]
Italy /Europe	Type I	*Ped. pentosaceus*, *Lat. curvatus*, *Lvb. brevis*, *Lim. fermentum*, *Lpb. plantarum*, *Ped. acidilactici*	*W. anomalus*, *P. fermentans*, *C. lusitaniae*, *S. cerevisiae*	[71]
France /Europe	Type I	*F. sanfranciscensis*, *Co. paralimentarius*, *Lvb. brevis*	*S. cerevisiae*, *K. humilis*, *K. bulderi*	[72]
Italy/Europe	Type I	*Lat. curvatus*, *F. sanfranciscensis*, *Lc. citreum*, *Lc. mesenteroides*, *Lc. pseudomesenteroides*, *Ped. pentosaceus*, *Lev. acidifarinae*	*C. humilis*, *T. delbrueckii*, *S. cerevisiae*, *K. marxianus*	[8]
Italy/Europe	Type I	*F. sanfranciscensis*	*C. milleri*, *S. cerevisiae*	[73]
Belgium/Europe	Type II	*Lim. fermentum*, *Lpb. plantarum*, *Lvb. brevis*, *W. confusa*, *Ped. pentosaceus*	*S. cerevisiae*	[74]
Belgium/Europe	Type I	*F. fructivorans*, *Lpb. plantarum*, *Lim. reuteri*, *Lb. delbrueckii*, *Lc. spp.*, *Weisella*	*C. humilis*, *S. cerevisiae*, *K. sp*, *P. kudriavzevii*	[75]
	Type II	*Lpb. plantarum*, *Lc. spp.*, *Lim. reuteri*, *Lb. delbrueckii*	*S. cerevisiae*, *W. anomalus*, *S. bayanus*, *T. delbrueckii*	
Italy/Europe	Type I	*F. sanfranciscensis*, *W. cibaria*, *Lpb. plantarum*, *Lim. reuteri*, *Lim. pontis*	*S. cerevisiae*, *K. exigua*	[76]

*LAB*: *Lc.*: *Leuconostoc*, *L.*: *Lactococcus*, *W.*: *Weisella*, *Lpb.*: *Lactiplantibacillus*, *Lvb.*: *Levilactobacillus*, *Lat.*: *Latilactobacillus*, *Ped.*: *Pediococcus*, *Flb.*: *Furfulactobacillus*, *F.*: *Fructolactobacillus*, *Lim.*: *Limosibacillus*, *Lcb.*: *Lacticaseibacillus*, *Co.*: *Companilactobacillus*, *Lev.*: *Levilactobacillus*, *E.*: *Enteroccoccus*, *Len.*: *Lentilactobacillus. Yeasts*: *W.*: *Wickerhamomyces*, *K.*: *Kasachstania*, *S.*: *Saccharomyces*, *C.*: *Candida*, *T.*: *Torulaspora*, *P.*: *Pichia*, *R.*: *Rhodotorula*, *H.*: *Hyphopichia*. * Sourdough Type I and II are explained in Section 4.2.

**Table 3 microorganisms-11-00109-t003:** Starter cultures for sourdough from different food matrices.

Food Matrix Used	Microorganisms	Inoculum Size	Main Results	References
Yogurt	*S. thermophilus* and *Lb. delbrueckii* subsp. *bulgaricus*	40% (*w*/*w*)	Enhanced bread has bread crumb softness, retarded staling, and increased antioxidant activity compared with yeast-sourdough	[16]
Corn bran	Two consortia: (1) *K. unispora + W. cibaria+ Ped. pentosaceus* and (2) *S. cerevisiae (commercial) + W. cibaria + Ped. pentosaceus*	6 log UFC/mL for all microorganisms except for *W. cibaria*	Spelt-sourdough bread obtained with the consortium (2) had a superior crumb texture that was maintained during five days of storage and has better consumer acceptation. Both consortia improved shelf life by preventing the growth of common cereal-contaminating fungi	[71]
Coconut water kefir	*Lim. fermentum* with and without yeast *Lpb. plantarum* with and without yeast	4.90 and 8.30 log UFC/mL 5.00 and 9.69 log UFC/mL	Sourdough bread inoculated with *Lpb. plantarum* at 9.60 log CFU/mL without yeast and fermented during 24 h showed a higher concentration of organic acids and amino acid, and better quality in terms of taste, shelf life, and texture	[133]
Cocoa bean fermentation, fermented sausage and water kefir	*Lim. fermentum* IMDO 222 (cocoa bean fermentation) *Lat. sakei* CTC 494 (fermented sausage) *Acetobacter pasteurianus* IMDO 386B and *Gluconobacter oxydans* IMDO A845	6–7 log UFC/mL of wheat flour-water mixture	*Lim. fermentum* IMDO 222 from cocoa bean fermentation and *Lat. sakei* CTC 494 from fermented sausage were potential starters for sourdough, as well AAB strains (*A. pasteurianus* IMDO 386B and *Gluconobacter oxydans* IMDO A845), both strains from cocoa bean fermentation), due to their competitiveness in the dough.	[14]
Water (WKG1, WKG2) and milk (MKG) kefir grains	*Len. kefiri* and *P. acidilactici* strains isolated from MKG	0.20% (*w*/*w*)	Using WKG2 as starter for sourdough in liquid and solid fermentation was exhibited a higher concentration of organic acids, flavonoids, and polyphenolic compounds with antioxidant and antifungal properties.	[18]
Pear and orange	*Lvb. brevis*, *Lpb. plantarum*, *Flb. rossiae,* and *S. cerevisiae*	200 g of fruit	The use of pear and orange as sourdough starters significantly decreased bread’s pH, acidity, and gas production, and increased free amino acids (FAA) content and gas holding capacity. Moreover, compared to the use of orange as starter, pear can achieve acidic conditions that are more suitable for the good performance of LAB and yeast during fermentation, resulting in a bread with a higher specific volume and a softer crumb.	[64]
Kimchi	*Lc. citreum* and *W. koreensis*	6 log CFU/g dough	The bread prepared with sourdough inoculated with kimchi LAB strains had significant effect on texture and could lead to an extended shelf life, by delaying bread staling and microbial spoilage.	[134]

## Data Availability

Not applicable.

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
