# Peer review of "Exploiting the Native Microorganisms from Different Food Matrices to Formulate Starter Cultures for Sourdough Bread Production"

_microorganisms, 2022, doi:10.3390/microorganisms11010109_

Round 1

Reviewer 1 Report

The manuscript written by Hernandez-Parada et al., is a review focused on the microbiota of sourdough bread.

The text is well written, the structure makes the reading very comprehensible and smooth; the references are recent.

There are just a few observations to do:

Check in the text the line spacing: it is not uniform

Line 37: yeasts

Line 68: this sentence is already reported at line 34

In table 2 the name of microorganisms should be written in full, since in the text not all the species are cited. Moreover, the authors should indicate that “type 1 and type 2” are explained in the paragraph 4.2

Fermentation time: is there an optimal fermentation time?

Environment: Are some environments better than others?

Are there preferred or more commonly used starter species for type II?

In my opinion, since this is a review, a short paragraph should be added where the organic acids, volatile compounds, polyphenols are described in order to help the reader better understand the influence of different species on these secondary metabolites.

After these minor revisions, the manuscript can be published in Microorganisms

Reviewer 2 Report

In my opinion, the review presented is current and interesting and provides a detailed picture of the subject investigated.

The article presents the topic exhaustively, giving a look at various aspects related to sourdough. In addition to general sourdough information, it provides insight into an emerging topic such as the use of different starter cultures from various food sources, from wheat flour to starter cultures. In my opinion, this review finds space in the literature, precisely because of its completeness of information.

Reviewer 3 Report

I recommend changing the title, especially the term "microbiota" because in this context it is not correct.

I recommend careful English proofreading. Throughout the text, there are many words repeated in the same sentence. For instance, “It is essential to consider the sourdough fermentation temperature, the refeeding temperature, and the temperature at which the sourdough is added during the baking process”

Lines 102 - 103 make references to figure 1 (wheat grain structure) and mention that the grain is divided into three main regions. However, figure 1 emphasizes only the pericarp and grain layers. I recommend changing the figure, perhaps using arrows indicating where the germ, endosperm, and pericarp layers are. Also, in the grain between brackets, other layers are specified, but it is not clear in the figure nor in the text which part of the grain they represent. This makes the reading a little confusing and difficult to understand.

Between lines, 103 - 107 is showing an estimate of how much each layer of the grain represents in percentage, but the sum result of these values equals 101%. Please review and adjust these values.

Please review the numbering of the tables and citations referring to them in the text.

Table 1 provides the protein composition of wheat flour. However, only 1 reference was used as a base. Couldn't this concentration vary according to climate conditions, crops, and varieties? I believe it would be more appropriate to use other references and make an average and standard deviation of these values.

I recommend joining the topics 4 Sourdough, 4.1 Sourdough Fermentation, and 4.1.2. Biochemical Transformations During Sourdough Fermentation, as there is a lot of repeated information.

Redundant information in Line 210 as the function of LAB had already been mentioned in the previous sentence. Please restructure the sentence "LAB species contribute to the dough acidification process",

In topic 4.1.1. Lactic Acid Bacteria (LAB), I recommend discussing a little more about starch metabolism by LAB; some studies have already described and characterized strains with aminolytic phenotypes.

Linhas 244-245: Por favor, adicionar referência após a frase:” However, yeasts have been shown to produce essential amino acids to facilitate LAB growth in co-cultures”.

I recommend removing the "Wheat Flour Type" topic or using this information in the "3. Wheat Flour" topic. Also, lines 283-287 are unclear and the topic overall has many redundancies.

I recommend a more detailed discussion on the topic "Environment" since the information brought by the authors (i.e. that the environment influences the microbiota) is common knowledge. A study by Comasio et al. is cited, perhaps using the information from this study (and other studies) can enrich the article. 

More information is repeated in the topic "4.2. Classification of Sourdough". For example, in lines 339 -342 some factors that influence fermentation are mentioned. However, the previous topics talked about these factors individually. Remove this topic.

I believe you could better explore item 5.1. Traditional Starter Cultures. It is only mentioned if there was a positive or negative result and it is not discussed why these results occurred. Adaptation of the microorganisms? Temperature? Etc

Please change the title of topic 5.2.

The use of alternative starter cultures may seem interesting because these fermentations can result in bread with unique sensory profiles. However, when we think about industrial production and/or process standardization, i.e. reproducibility of the sensory profile, it is practically unfeasible to use these fermentation methods. I think the authors could bring a critical analysis regarding these problems, perhaps in the conclusion or at the end of topic 5.2.

Round 2

Reviewer 3 Report

All suggestions made to the authors were answered or justified clearly. The changes in the text made it a little smoother to read. The changes in figure 1 and the insertion of the figure 3 significantly helped the understanding of the text. However, I believe that the title of the article could still be improved, but I leave it up to the authors to decide this.

To assist here follows a suggested title: "Exploiting the native microbiota from different food matrices to formulate starter cultures for sourdough bread production"

Author Response

At the reviewer's suggestion, the article's title was modified. In addition, we have included minor edits to improve readability.